# High speed underwater hydrogel robots with programmable motions powered by light

Chujun Ni[1], Di Chen[2], Xin Wen[1], Binjie Jin[1], Yi He [1], Tao Xie [1,2] & Qian Zhao [1,2] ✉

Stimuli-responsive shape-changing hydrogels are attractive candidates for use as underwater soft robots. The bottleneck lies in the low actuation speed inherently limited by the water diffusion between hydrogels and their surrounding environment. In addition, accessing complex motions is restricted by the material fabrication methods. Here we report a hitherto unknown mechanism to achieve high-speed and programmable actuations for a disulfide crosslinked thermally responsive hydrogel. The dynamic photo-activated disulfide bond exchange allows photo-mechanical programming to introduce spatio-selective network anisotropy. This gives rise to an actuation behavior dominated by thermally driven conformation change of the locally oriented polymer chains instead of the common mass-diffusion-based mechanism. With the incorporation of photothermal fillers, light-powered oscillation at frequencies as high as 1.7 Hz is realized. This, coupled with the versatility of the programming, allows access to robots with diverse high-speed motions including continuous swimming, step-wise walking, and rotating.

Robots driven by soft actuators[1–7] are gaining increasing attention due to their ability to accomplish tasks in complex scenarios that are challenging for conventional rigid robots. Amongst various soft actuators, hydrogels stand out in their water-rich nature, tissue-like softness, and molecular designability[8–10] toward diverse stimulation methods, including temperature[11–14], pH[15,16], light[17–19], magnetic field[20], and electricity[21,22]. The actuation behaviors of hydrogels typically originate from the stimuli-triggered swelling and deswelling[23–25]. Consequently, the actuation speed is limited by the water diffusion kinetics. For monolithic hydrogels, each actuation cycle typically requires minutes to hours to complete depending on the sample dimensions[26–33].

Shortening the water diffusion path is an effective approach to accelerate the actuation. This can be done by reducing the film thickness[34], making hydrogel fibers[11,12,35,36] and introducing porosity[37,38]. For instance, an organohydrogel composed of hydrogel bundles and oleophilic confinement allows shortening the water pathway to realize high frequency (0.1 Hz) actuation[12]. However, either dimensional restrictions or the associated processing requirements compromise the freedom for designing robots with complex motions, that is, achieving nonlinear actuations (e.g., bending or twisting) is challenging. Other limitations are, for instance, a significant reduction of volumetric actuation output in the case of porous hydrogels. For nonporous monolithic hydrogels, a unique mechanism to achieve rapid actuation is accomplished by incorporating cofacially oriented electrolyte nanosheets in a thermo-responsive hydrogel[39]. The electrostatic repulsion between the nanosheets can be directly modulated by temperature, resulting in actuation. Since the mechanism is independent of water diffusion, the actuation can occur in 1 s. Despite the elegance, however, the nanosheet alignment relies on a magnetic field. Designing complex motions would demand access to sophisticated magnetic fields that are challenging to create.

Irrespective of the speed, converting the natural isotropic volume expansion/contraction of responsive hydrogels to complex actuation motions is also key. A common strategy is to introduce spatial material heterogeneity[40–43]. For hydrogels with spatial variation of the

---

[1]State Key Laboratory of Chemical Engineering, College of Chemical and Biological Engineering, Zhejiang University, 310058 Hangzhou, China. [2]Ningbo innovation center, Zhejiang University, 315100 Ningbo, China. ✉e-mail: qianzhao@zju.edu.cn

crosslinking densities, for instance, intricate motions are realized through the localized uneven swelling/deswelling. Typically, however, the material heterogeneity consequently the actuation mode is determined/fixed during the material synthesis step and cannot be altered afterward, namely, the motions cannot be repeatedly reprogrammed. By introducing long aliphatic side chains in a thermoreversible hydrogel, reprogrammability becomes feasible via the additional crystalline/melting transition[44]. However, completing an actuation cycle requires tens of minutes dominated by the slow cooling-induced actuation step. Specifically, the collapsed PNIPAM chains tend to aggregate around the hydrophobic domains, and the conformation change upon cooling would be slow since the strong hydrophobic interaction remarkably retard the chains from reswelling.

Designing a high-performance hydrogel robot demands simultaneously achieving fast actuation and programmable complex motions, a task that has been proven challenging despite the significant progress in the past. Here we report our successful effort in achieving this goal with a thermo-responsive hydrogel crosslinked by disulfide bonds. The dynamic exchange nature of the disulfide bond, which gains from the UV-triggered break-recouple process following radical mechanism[45], allows photo-mechanical programming to introduce spatio-selective network anisotropy. This gives rise to 3D complex motions and, most importantly, an actuation mechanism dominated by thermally controlled chain conformation change instead of water diffusion. Coupling this unique mechanism with photothermal heating, light-powered high-speed hydrogel robots with diverse complex motions are realized.

## Results

Our discovery of the fast and reversible actuation behavior for a nonporous monolithic hydrogel is accidental. We recently developed a dynamic crosslinked polyacrylamide hydrogel containing both disulfide and ionic bonds[46]. The hydrogel exhibits temporal-programmable shape-shifting behavior and its permanent shape can be reconfigured. However, its shape-shifting is limited to one-way (irreversible) shape memory, not the reversible actuation required for

building soft robots. In this work, we replace the non-responsive polyacrylamide backbone with a thermo-responsive poly(N-isopropylacrylamide) (PNIPAM) in the hydrogel network. To our pleasant surprise, the resulting hydrogel not only exhibits programmable reversible actuation, but its actuation can occur within seconds.

The precursors of our hydrogel are shown in Fig. 1a. In its aqueous solution containing poly(vinyl alcohol) (PVA), the NIPAM monomer is polymerized/crosslinked with a disulfide crosslinker (BISS). The hydrogel is further crosslinked with aluminum ions. After the sample preparation, the hydrogel is subjected to a photo-mechanical programming process, in which it is deformed under an external force and irradiated with ultraviolet (UV) light. The UV exposure triggers the disulfide exchange and the original shape is reconfigured into a new shape (Fig. 1b). After the programming, the hydrogel can perform reversible actuation at the deformed regions (see Fig. 1b and Supplementary Movie 1). This behavior is different from the temperature-triggered isotropic volume change of the hydrogel without programming (Supplementary Fig. 1).

The underlying mechanism of such a reversible actuation behavior is intuitively shown in Fig. 1c. The as-synthesized hydrogel is in its isotropic state corresponding to no chain orientation. Through photo-mechanical programming, the shape change is attributed to the rearrangement of the network topology based on disulfide bond exchange. When the sample is deformed into a certain strain, the polymer chains are stretched, and the entropy of the entire network decreases. Upon activation of the disulfide exchange via UV exposure when retaining the deformation, the polymer network can dissipate the stored energy through the topology rearrangement thus the system entropy increases again. As a result, the polymer network will not recover to the original state upon removal of the external force, and thus, the macroscopic shape changes after the UV programming[46]. We note that the bond exchange decreases along the thickness due to the light attenuation, resulting in gradient stress relaxation. As such, the hydrogel in Fig. 1c reconfigures its permanent shape that bends away from the UV exposure side after releasing the external force. At the same time, the polymer chains in the network become partially

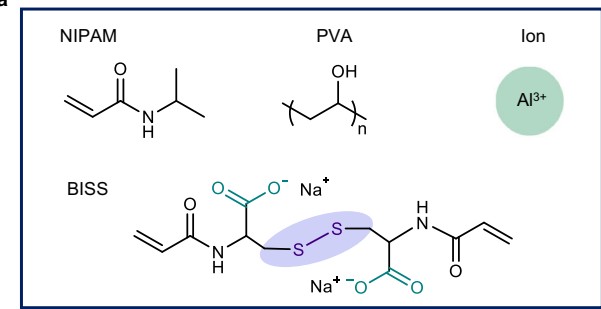

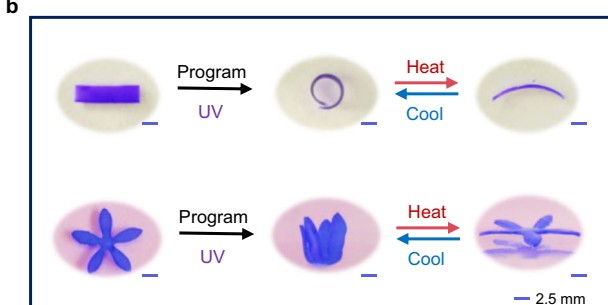

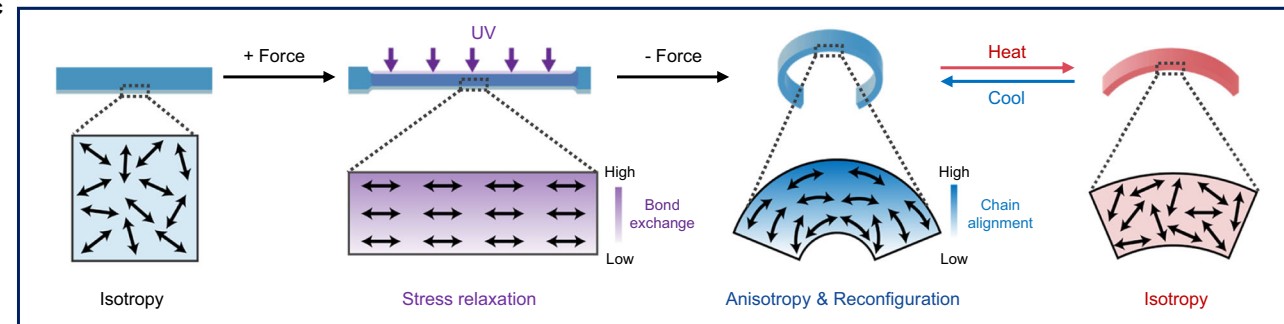

**Fig. 1 | Mechanism and demonstrations of the anisotropic hydrogel actuator. a** Precursors of the hydrogel. **b** Photographs showing the programming step and the thermo-reversible actuation of the anisotropic hydrogels. The scale bars are 2.5 mm. **c** Molecular mechanism of the shape reconfiguration and actuation.

oriented[47,48] due to the stress distribution, which was supported by the polarized optical microscopic images (Supplementary Fig. 2). After programming, the aligned PNIPAM chains collapse into their isotropic state along the orientation direction when heated above LCST and the process reverses upon cooling. Accordingly, reversible actuation is realized by a temperature switch. Herein a lamination method is used to verify the gradient in the chain orientation. Three identical hydrogel sheets with a thickness of 0.1 mm were overlayed, and the integrated sample was programmed (folded and irradiated) through the same approach as the common non-laminated sample. The angles of the three sheets after the programming were respectively 41°, 95° and 115° from top to bottom (Supplementary Fig. 3), indicating a decreasing trend of shape retention because of the light attenuation along the thickness direction. The experimental results support the proposed mechanism of the gradient on chain orientation, implying the existence of internal stress within the sample.

Our hydrogel meets two basic requirements for reversible actuation[49]: a mechanism to permanently align the polymer chain for molecular anisotropy (dynamic bond exchange of BISS) and a mechanism to switch the polymer conformation for actuation (LCST transition of PNIPAM). Besides meeting these two prerequisites, ion crosslinking and PVA are also indispensable for this system. Aluminum ions are used to improve the mechanical properties of the material, otherwise, the hydrogel cannot sustain large mechanical deformation during programming (Supplementary Fig. 4). The introduction of PVA can build up water channels to accelerate water transportation[50], avoiding the generation of blisters in pure PNIPAM hydrogel as shown in Supplementary Fig. 5. The PVA helps to ensure smooth and stable shape transformations during the actuation cycles.

Quantitative investigations were conducted to optimize the actuation factors by tuning the material compositions and programming parameters. A bending model was chosen to quantify the shape reconfiguration and the actuation amplitude after programming (Fig. 2a and Supplementary Fig. 6). Here, the extent of shape reconfiguration is defined as shape retention, which is (180°-

$\theta_0$)/180° × 100%, where $\theta_0$ represents the initial angle of the hydrogel after programming. The actuation amplitude is calculated as the actuation angle ($\theta_{max}$-$\theta_0$), where $\theta_{max}$ corresponds to the maximum angle of the actuation. The concentration of the disulfide bonds is expected to be crucial to the photo-mechanical programming. Accordingly, a series of hydrogel samples with different BISS contents were synthesized. Before complexation with aluminum ions, the hydrogel showed no thermal responsiveness because of the high hydrophilicity of the ionized carboxyl groups in BISS. After complexation, the carboxyl groups became less hydrophilic providing decreasing LSCT from 34 °C to 27 °C with the BISS content (Supplementary Fig. 7). All the samples show good mechanical properties with strains at break greater than 100% (Supplementary Fig. 8). Figure 2b shows that the shape retention changes non-monotonically with the BISS content. When the BISS content was low, there were only a few exchangeable crosslinking points. The shape retention was consequently low (approximately 20%) and the actuation angle is quite small due to the low degree of chain orientation. Oppositely, if BISS content is too high (larger than 60 wt%), the densely crosslinked hydrogel network would also impede the dynamic bond exchange. The balance of the two above factors is such that a maximum value of shape retention was achieved when 40 wt% BISS was incorporated into the network. The actuation angle follows a similar trend with that of the shape retention. This is consistent with the fact that both the actuation and shape retention share the same root in the bond exchange-induced molecular chain alignment. Hereafter, the optimal BISS content of 40 wt% was chosen for further investigation. The UV irradiation time during photo-mechanical programming is another parameter that controls the extent of the disulfide exchange. The shape retention increases with the irradiation time and reaches a plateau value of approximately 80% after 300 s. Likewise, the actuation angle gradually increases to a maximum value of 90° at 300 s (Fig. 2c). Hence the irradiation time was set as 300 s to promote maximum performances in both the shape retention and actuation.

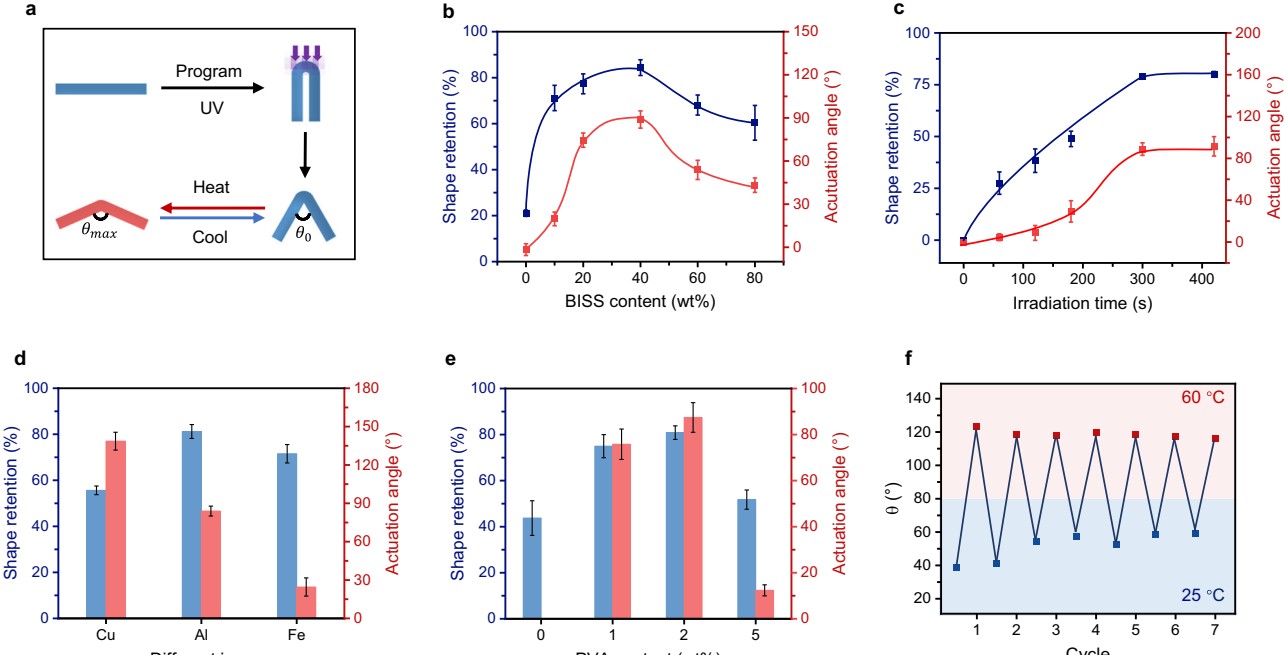

**Fig. 2 | Factors affecting the shape reconfiguration and the actuation. a** Scheme of the bending model for the performance evaluation. **b** The impact of BISS contents. The irradiation time was 300 s. **c** The effect of irradiation time. **d** The influence of ion complexation. **e** The function of PVA addition. **f** The actuation cyclability of the hydrogel. The actuation temperatures were set to 60 °C and 25 °C respectively throughout the experiments. All the error bars correspond to s.d. (*n* = 5).

The influence of different multivalence metal ions ($Fe^{3+}$, $Al^{3+}$, and $Cu^{2+}$) was subsequently studied. The mechanical enhancement follows the order of $Fe^{3+}>Al^{3+}>Cu^{2+}$ (Supplementary Fig. 9). However, the actuation angle varies in an opposite way ($Fe^{3+}<Al^{3+}<Cu^{2+}$, Fig. 2d) since the deformation resistance in more rigid materials is also higher. $Al^{3+}$ was picked to balance the contradictory performance. The effect of PVA addition is presented in Fig. 2e. Addition of 2 wt% PVA results in the largest actuation angle and is chosen hereafter. After optimizing the chemical composition and programming time, the actuation durability of the hydrogel was evaluated (Fig. 2f). The actuation experiences a slight decrease after the first cycle due to the relaxation of the internal stress which partially eliminates the programmed network orientation. In subsequent cycles, however, the actuation angle keeps at a relatively constant value of approximately 75°.

The photo-mechanical programming offers unusual freedom for customizing the actuation modes by altering the external deformation force during the process. Figure 3a illustrates that two distinct actuation modes can be realized from an identical sample, with the only difference being the programming force applied. Figure 3b demonstrates chronological programming. A flat hydrogel palm was first programmed into an "ok" gesture, in which only two fingers show actuation. In a subsequent step, the other three fingers are also programmed, leading to an overall fist shape for which all five fingers can undergo actuation. Taking advantage of the spatio-selective nature of the photo exposure, actuation regions of a uniformly deformed sample can be locally defined (Fig. 3c), which further highlights the versatility and freedom of the reconfigurable actuations.

In the above-mentioned sections, we have presented the actuation performances of the hydrogel including the actuation amplitude and the photo-programmable actuation modes. More importantly, the actuation speed of our hydrogel is quite fast compared with typical osmotic-driven hydrogel actuators. Hereafter, its response kinetics was investigated and compared with a standard bilayer hydrogel actuator (Fig. 4). The bilayer hydrogel is composed of a thermo-responsive layer (identical to the above hydrogel) and a non-responsive passive polyacrylamide hydrogel layer. Upon heating/cooling, the mismatch of the volume change in the two layers leads to a bending action (Fig. 4a). Upon heating and cooling, its actuation kinetics as reflected in the curvature variation is consistent with the deswelling/swelling kinetics of the active layer. This suggests that the actuation speed is dominated by the mass diffusion process during swelling (Fig. 4a). Since the mass diffusion is slow, it takes tens of minutes for the bilayer to reach its equilibrium bending. In contrast, the actuation mechanism of our programmable hydrogel actuator is mainly dependent on the conformation reconfiguration of the polymer chains (Fig. 4b). The oriented chains in the programmed sample undergoes contraction upon heating and recovery upon cooling along the programming direction, with the intrinsic orientation gradient along the thickness direction further amplifying the nonlinear actuation. Consequently, the actuation is dominated by heat conduction rather than water diffusion.

The shape-shifting kinetics of our actuator in Fig. 4b provides strong support to the proposed mechanism. The actuator completes approximately 80% shape-shifting in 10 s, which is tens of times faster than the change in the swelling ratio. This trend is consistent with the theoretical kinetics, where the calculated time for accomplishing heat transport and mass diffusion is in the range of $10^{-1}$-$10^{1}$ s and $10^{3}$-$10^{5}$ s, respectively (Supplementary Note). Further experiments carried out in oil environment (instead of water) also support our hypothesis (Fig. 4c). The programmed sample exhibits fast shape-shifting when immersed in hot oil for 10 s and recovers to its original shape in cold oil in 30 s. By comparison, a non-programmed sample does not show obvious volume shrinkage until heating for 10 min, which means that the mass diffusion contributes little to the actuation in the short timescale.

For a more straightforward comparison, the real-time actuation degree, defined as the ratio between the deformation at a given time

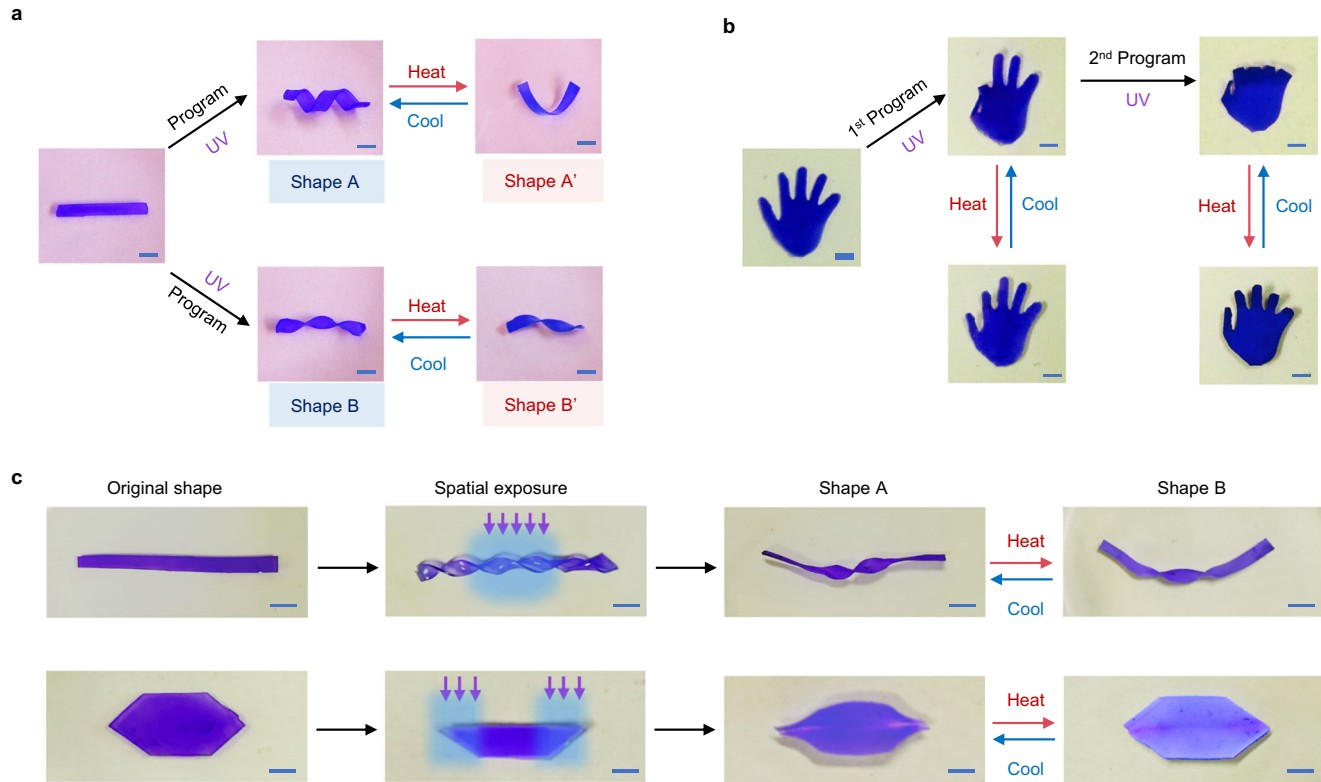

Fig. 3 | Reversible actuations of the photo-mechanically programmed hydrogel. a Two actuation modes enabled by applying two different programming forces on identical samples. b Chronological programming. c Spatio-selective programming of hydrogels for regional selective actuations. All the scale bars are 5 mm.

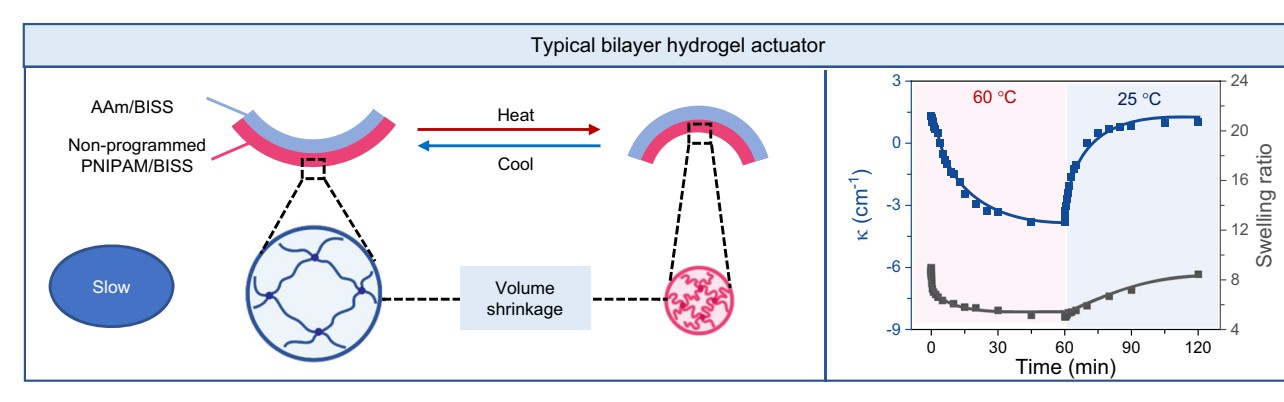

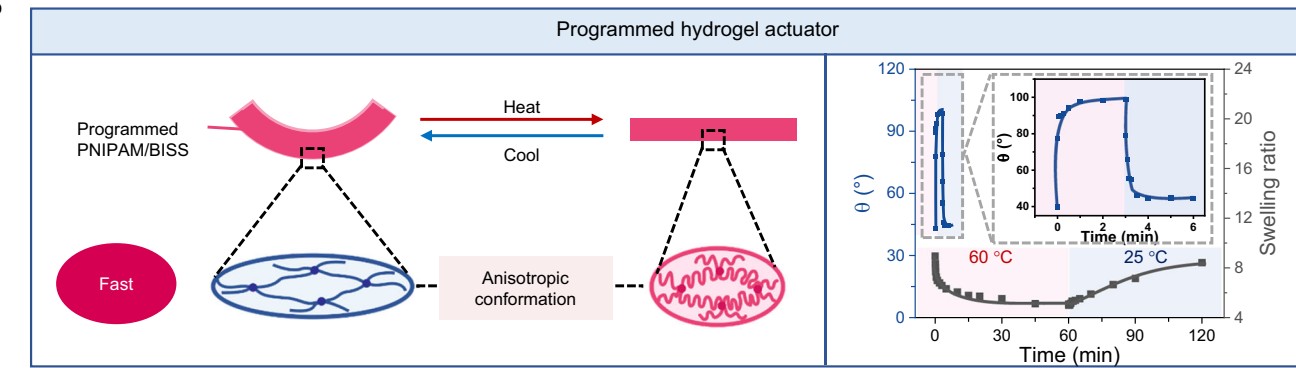

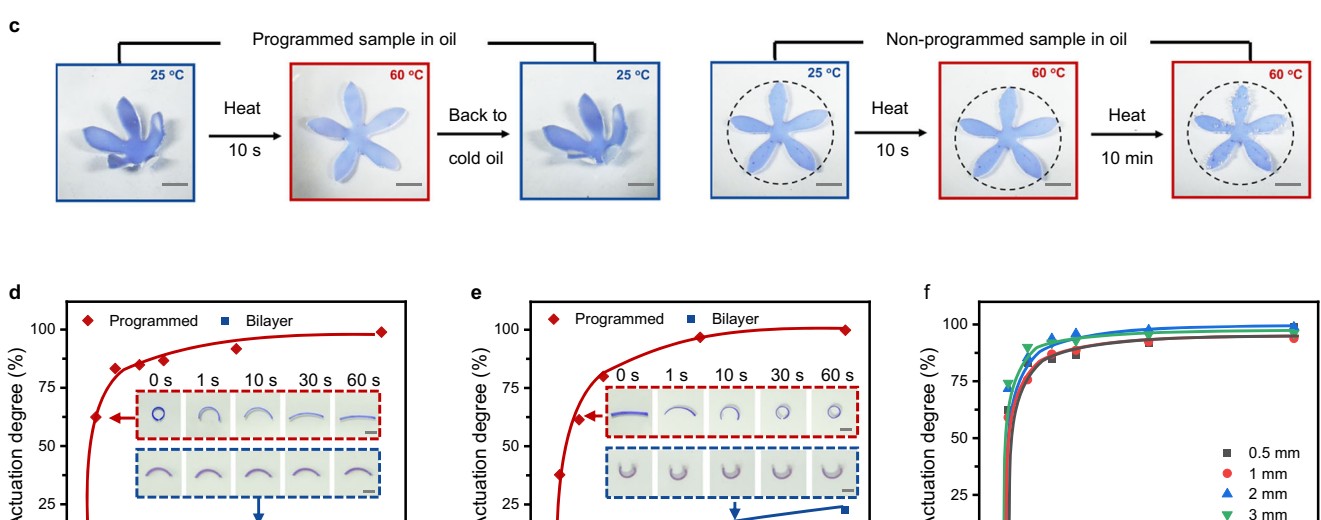

**Fig. 4 | Comparison of the actuation kinetics between our programmable hydrogel and a bilayer reference hydrogel. a** Mechanism of the mass-diffusion-controlled actuation of a typical bilayer actuator and its actuation and swelling kinetics. **b** Mechanism of the heat-diffusion-controlled actuation of the anisotropic hydrogel actuator and its actuation and swelling kinetics. **c** Comparison of the actuation of programmed and non-programmed samples in oil. The scale bars are 2.5 mm. **d** Actuation kinetics of the two actuators upon heating at 60 °C. The scale bars are 5 mm. **e** Actuation kinetics of the two actuators upon cooling at 25 °C. The scale bars are 5 mm. **f** Actuation kinetics of hydrogel samples with different thicknesses.

and the maximum deformation, is extracted from Fig. 4a, b. More specifically, the actuation degree of bilayer actuator/programmed actuator is defined as $(x_t - x_0)/(x_{max} - x_0) \times 100\%$, where $x_t$, $x_0$, and $x_{max}$ represents the real-time, initial, and maximum actuation values, respectively. For the bilayer actuator, $x$ is the curvature value $\kappa$. For the programmed actuator, $x$ is the angle value $\theta$. Figure 4e shows that the actuation is nearly complete in 30 s of heating for our programmable

actuator. Under the same condition, the actuation degree is only 13% for a classical bilayer actuator. Likewise, cooling-induced actuation for our programmable actuator is also much faster than the bilayer reference hydrogel, with their actuation degrees reaching approximately 100% and 23% respectively in 30 s. The inset photos and Supplementary Movie 2 further illustrate the large contrast between the shape-changing kinetics of the two types of actuators. Because the

actuation of our hydrogel is no longer governed by mass diffusion, its actuation kinetics is insensitive to sample geometry as reflected quantitatively in thickness (Fig. 4f and Supplementary Fig. 10). This is because the timescale of heat transport can be neglected in the investigated thickness range (0.5 mm to 3 mm). The geometric insensitivity unleashes the freedom to design complex yet highly controllable motions. This feature stands in sharp contrast to conventional hydrogel actuators with intrinsic geometric sensitivity due to its mass transport mechanism.

Apart from the distinct mechanism, the hydrogels also present some particular advantages in actuation performances in comparison with existing actuation materials. Actuation of two-way shape memory polymers relies on the melting/formation of crystalline domains which exhibit large thermo-hysteresis within a heating-cooling cycle. Therefore, the actuation temperature range is large, greatly compromising the actuation speed. For liquid crystal elastomers, achieving the actuation programmability and reducing their actuation temperature to around body temperature are both very challenging, since they require highly sophisticated chemical design. In contrast, the presented hydrogel system provides a facile and inspiring way to realize the programmability in a very mild condition as well as fast actuation in a narrow temperature variation. Moreover, the water-abundant nature of hydrogels promises their biocompatibility, tissue-like modulus and unique mass-transfer property (for drug delivery), which are desired for many biomedical-related applications.

Although our hydrogel displays fast actuation capability, changing the entire surrounding water temperature rapidly is practically infeasible to realize the high-frequency actuation required for robotic applications. We resort to endogenous photothermal heating by doping carbon particles into the hydrogel. The doping did not result in remarkable alteration in the mechanical properties of the hydrogel (Supplementary Fig. 11), but introduced a larger stress gradient along the thickness due to the enhanced light absorption of carbon doping (Supplementary Fig. 12). The shape retention reduced to $53.2 \pm 2.2\%$, and the actuation angle slightly decreased to $74.7 \pm 4.2°$ (Supplementary Fig. 13). On this basis, remote NIR-irradiation control is further investigated. Notably, when the distance between the NIR source and the hydrogel sample is fixed at 4 cm, the irradiation area is 22 mm² which fully covers the actuation region of our hydrogel samples (Supplementary Fig. 14). The photothermal efficiency at this condition is reported in Supplementary Fig. 15. Under near-infrared (NIR) irradiation, the sample temperature rises above the LCST within 1 s and reaches its equilibrium of 47 °C in 3 s. After removal of the light, it cools naturally by the surrounding water to room temperature in 10 s. Under periodic irradiation, a programmed hydrogel is capable of ultrafast oscillations at frequencies from 0.3 Hz to 1.7 Hz (Fig. 5a and Supplementary Movie 3). In comparison with other thermo-sensitive PNIPAM hydrogel actuators reported in the literature, our material simultaneously provides good mechanical rigidity (modulus) and fast actuation (Fig. 5b). These are two aspects key to designing untethered free-standing robots. Of course, our hydrogel actuator can also be repeatedly (re)-programmed into different shapes with diverse actuation modes, a characteristic not offered by common hydrogel actuators (Fig. 5c). Taking advantage of the spatio-selective irradiation, a high-speed hydrogel swimmer is fabricated with its continuous movement driven by the fast oscillatory swing of the limb (Supplementary Movie 4). After reprogramming, the swimmer is turned into a crawler with step-wise crawling by alternate irradiation on its two legs (Supplementary Movie 5). The crawler can be further programmed into a rotator that shows an anticlockwise rotation when the light is irradiated parodically at one end of the limb (Supplementary Movie 6). Overall, the ability to program a single hydrogel into high-speed underwater robots with diverse motions goes beyond currently known hydrogel actuators.

## Discussion

In this work, photo-responsive dynamic bonds were introduced into a thermo-responsive hydrogel network. Their bond exchange capability allows photo-mechanical programming of the macroscopic shape and microscopic network anisotropy. The latter gives rise to a unique actuation mechanism governed by the heat transport limited conformation change of the oriented chain segments, instead of mass diffusion limited swelling-deswelling for common hydrogel actuators. Consequently, the hydrogel exhibits an actuation speed that is three orders of magnitude higher than other hydrogels capable of large nonlinear actuations. In addition, a photothermal effect is integrated into the hydrogel by doping, which enables rapid heating by NIR light. The unique actuation capability, coupled with the shape programmability and remote light powering, allows the design of multi-modal high-speed underwater robots. Despite the desirable characteristics, there are two challenges related to the current hydrogel system. The first is the light powering. To achieve the desired motions, the NIR light is manually focused onto specific regions and has to move along with the robots, which is not easy to control. For future systems, one can incorporate the photothermal capability into specific regions of the material such that a flood light can trigger local actuation. The second is that, for high-speed motions, the speed is gained at the sacrifice of the actuation amplitude. This is because a full actuation cycle cannot be completed within the very short timescale inherently associated with the high speed. Although this does not negatively affect the functions of the demonstrated underwater robots, it would limit future opportunities that demand both high speed and large amplitude. Nevertheless, we believe that the unique actuation mechanism demonstrated in this work is useful for the future designing of advanced hydrogel devices.

## Methods

### Materials
$N$-isopropylacrylamide (NIPAM, ≥98%) was acquired from Macklin and recrystallized by $n$-hexane before use. $N,N'$-methylenebisacrylamide (BIS, ≥99%) was obtained from Sigma-Aldrich. Polyvinyl alcohol (PVA, $M_w = 89,000-98,000$, 99+% hydrolyzed) and carbon particles were bought from Aladdin. Acrylamide (AAm, ≥99%), tetramethylethylenediamine (TEMED, ≥99%), 2-hydroxy-4′-(2-hydroxyethoxy)−2-methylpropiophenone (I2959, ≥98%) were procured from J&K. L-cystine (≥99.5%) and acryloyl chloride (≥98%) were purchased from TCI. Aluminum sulfate octadecahydrate, copper chloride, ferric chloride, ether, hexane, sodium hydroxide, ammonium persulfate (APS), and methanol were purchased from Sinopharm Chemical Reagent Co., Ltd with AR purity.

### Synthesis of programmable hydrogels
Dynamic disulfide crosslinker $N,N'$-bis(acryloyl)-(l)-cystine salt (BISS) was synthesized according to the literature[46]. Then, 0.25 g NIPAM and 10−80 wt% BISS (relative to NIPAM) were dissolved in 2.5 mL PVA solution with 0−5 wt% concentration. Afterward, 50 μL 4 wt% APS aqueous solution and 10 μL TEMED were added to the solution. This precursor was immediately poured into the mold made of two glass sheets and a silicon spacer with 0.5 mm thickness, followed by polymerization at 4 °C for 24 h. The cured hydrogel was immersed in I2959 ionic solution containing 1 mg/mL I2959 and 0.01 M $Al^{3+}$/ $Fe^{3+}$/$Cu^{2+}$ for another 24 h to introduce photo-responsive bond exchange and ionic crosslinking. For photothermal responsive actuators, additional 0.005 g carbon particles with 150 μm diameter were added into the hydrogel precursor for the preparation. Unless otherwise noted, 40 wt % BISS, 2 wt% PVA, and $Al^{3+}$ were used to synthesize the hydrogels.

### Synthesis of bilayer hydrogels
For this, 0.25 g AAm, 0.1 g BISS, and 50 μL 4 wt% APS aqueous solution and 10 μL TEMED were dissolved in 2.5 mL DI water to form the PAAm

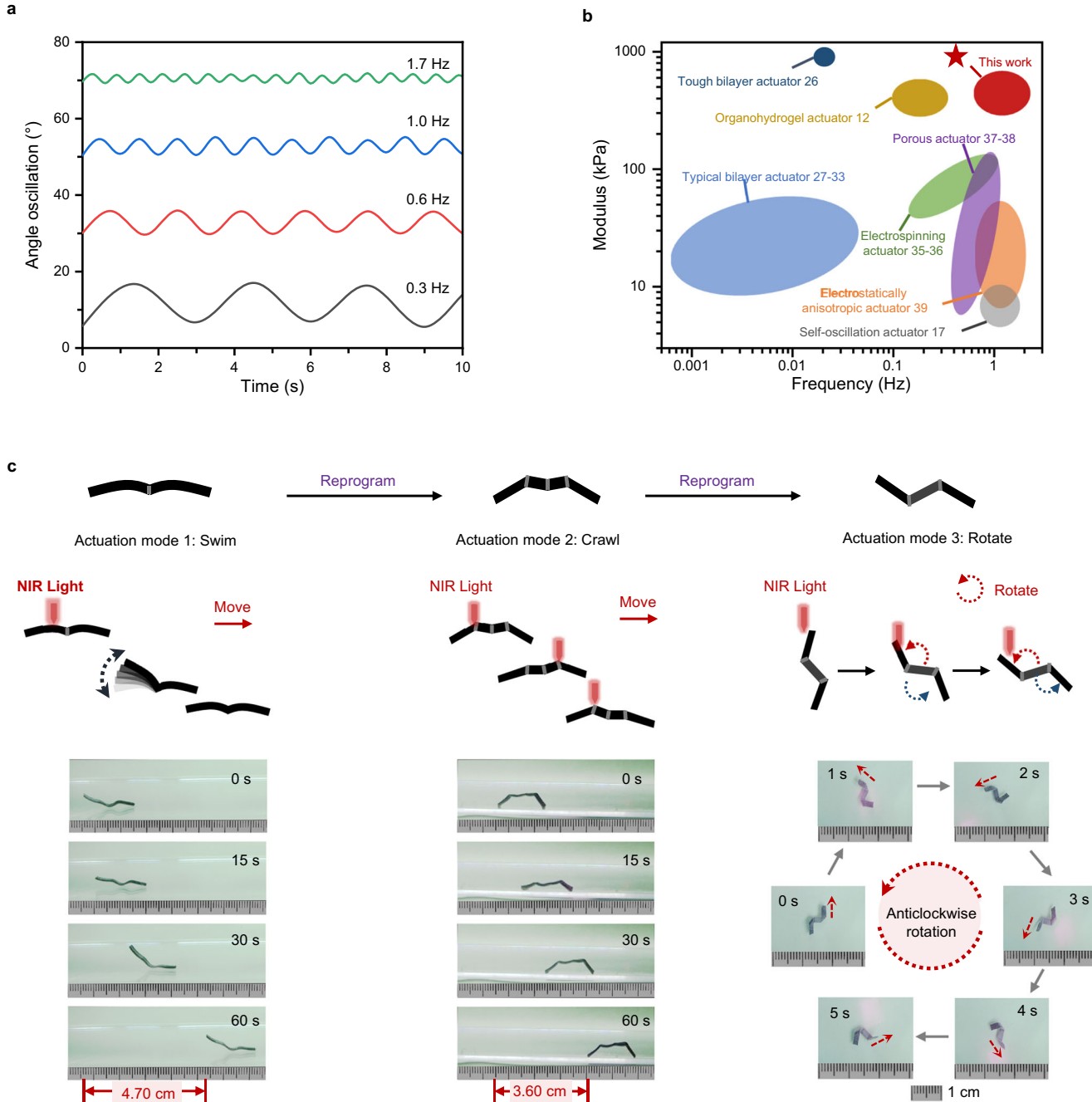

**Fig. 5 | Photothermally powered hydrogel robots with programmable multimodal actuations. a** Oscillatory actuation upon periodic irradiation varying from 0.3 Hz to 1.7 Hz. **b** Modulus and actuation frequency of the hydrogel actuators in comparison with the ones in the literature. **c** (Re)programmable multi-modal underwater robot from a single hydrogel sample.

hydrogel precursor. The solution was then poured into the mold and polymerized at 25 °C environment for 24 h to obtain the first layer. Then, the aforementioned PNIPAM hydrogel precursor solution was poured on top of the PAAm hydrogel layer and polymerized at 4 °C for 24 h. Each layer was controlled to 0.25 mm. The bilayer hydrogel was immersed in 0.01 M I2959 aluminum solution for 24 h to form subsequent ionic crosslinking. The synthesis and actuation process were shown in Supplementary Fig. 16. It is noted that the parameter of curvature $\kappa$ was chosen to reflect the shape-shifting kinetics (Fig. 4a). The $\kappa$ was set as positive at the cooling state and negative at the heating state. The deformation degree was calculated as $(\kappa_t - \kappa_0)/(\kappa_{max} - \kappa_0) \times 100\%$, where $\kappa_t, \kappa_0,$ and $\kappa_{max}$ represented the real-time, initial, and maximum curvature throughout the process respectively.

## Shape programming and actuation process

For programming, a $3 \times 10 \times 0.3$ mm hydrogel sample was bent into half and exposed to 80 mW/cm² UV light (CEL-HXF300, CEAU Light Co. Ltd.) as shown in Fig. 2a. The sample was folded into half and placed between two glass slides fixed by two clips (Supplementary Fig. 6). The irradiation time varied from 0 min to 7 min. Afterward, the sample was put back into the 25 °C aqueous environment without external force until equilibrium. The remaining angle of the sample was defined as $\theta_0$. The shape retention was calculated as $(180°-\theta_0)/180° \times 100\%$. When the sample was transferred to 60 °C aqueous environment until equilibrium, the angle of the sample was defined as $\theta_{max}$. The actuation angle $\theta$ was calculated as $\theta_{max} - \theta_0$, which was also chosen to reflect the shape-shifting kinetics in Fig. 4b. Each result was calculated from

five parallel samples. Unless otherwise noted, the irradiation time for the programming was fixed to 300 s. For spatial irradiation, the unexposed areas were shielded by tin foils.

### Swelling ratio
The weight of hydrogels in the fully swollen state and dried state were measured as $m_{wet}$ and $m_{dry}$, respectively. The swelling ratio was identified as $m_{wet}/m_{dry}$ calculated from five parallel samples.

### DSC characterization
LCST transition was tested by DSC measurements (TA Q200) with a ramp rate of 5 °C/min.

### Tensile tests
Tensile tests were carried out by Care Test IPBF-300 at a strain rate of 5 mm/min in an ambient water bath. The results were conducted for five parallel samples.

### Photothermal efficiency measurements
The hydrogel sample was irradiated by the Infrared light (LSR808HX, Lasever Inc.) with 45 W power and 808 nm wavelength. The temperature variation was monitored by a Fotric 237 thermometer. The distance between a sample to the light source was fixed at 4 cm.

### Polarized optical imaging
The polarized optical images were captured by a polarized optical microscope (ECLIPSE E600W POL) at room temperature.

## Data availability
Additional data are available from the corresponding author upon request.

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

## Acknowledgements
We thank the National Key R&D Program of China (No. 2022YFA1103500 to Q.Z.) and the National Natural Science Foundation of China (No. 52273112 to Q.Z., No. U20A6001 to Q.Z., No. 22105167 to D.C.). We thank Mrs. Li Xu and Mrs. Sudan Shen for the help of DSC and polarizing microscopy characterizations at the State Key Laboratory of Chemical Engineering (Zhejiang University).

## Author contributions
Q.Z., C.N. and D.C. conceived the concept. C.N. performed most of the experiments. X.W. assisted in conducting the experiments. D.C., B.J., Y.H., and T.X. analyzed the mechanism. The manuscript was written by C.N., T.X., and Q.Z. All authors discussed the results.

## Competing interests
The authors declare no competing interests.
