## [Peer review file · Nature Communications]

REVIEWER COMMENTS

Reviewer #1 (Remarks to the Author):

High actuation speed and reconfigurable actuation modes are two desired characteristics for soft hydrogel robots. Despite recent progresses, material systems exhibiting both are still under-explored. This manuscript presents a photothermally responsive hydrogel with rapid motions for which actuation modes can be programmed via dynamic exchange of the disulfide crosslinking. The actuation performances are impressive. In contrast to common mass-diffusion-based mechanism for previous hydrogel actuators, the manuscript demonstrated a unique mechanism for which the actuation relies mainly on conformational change of the anisotropic polymer chains after programming. Therefore, I would like to recommend its publication in Nature Communications after certain revisions by addressing the following issues.

1. It is not clear why the actuation speed of previous hydrogels with re-programmability (ref. 42) is slow. Please elaborate.
2. It is expected that there is a gradient on the chain orientation, as reflected in the shape retention. However, the evidence is somewhat insufficient. More investigations concerning the gradient should be conducted.
3. It is known that two-way shape memory polymers and liquid crystal elastomers can also provide programmable actuations. Apart from the distinct mechanism, advantages on material performances of the presented hydrogels should be discussed.
4. More characterizations (e.g. tensile tests) are required to show the material properties after doping the carbon particles.

Reviewer #2 (Remarks to the Author):

In this paper, the authors developed a soft underwater robot based on thermal-responsive hydrogel crosslinked by a disulfide crosslinker (BISS). Owing to the photo-responsive dynamic bonds in hydrogel network, the robot allows photo-mechanical programming to introduce diverse motion modes. The factors affecting the shape reconfiguration are explored and the optimal parameters for best actuation performance are chosen. Moreover, the authors also compared the speed of conventional actuation mechanism based on water diffusion and this actuation mechanism. Based on the conformation reconfiguration of the polymer chains, the hydrogel robot realizes fast actuation (1.7 Hz), with great potential for fabrication of advanced hydrogel devices in the future. In general, this work is interesting and clearly organized. However, there are still some details remaining to be improved before acceptance.

1. The authors introduced the actuation mechanism of this hydrogel robot and mentioned that the stress of a uniaxial stretched hydrogel sample will be relaxed through photo-mechanical programming via disulfide exchange. Why dose UV irradiation result in disulfide exchange? To make it clear, the reason should be added.
2. During shape reconfiguration, the authors mentioned that external force should be applied under UV exposure. The reviewer is curious about how to apply external force under UV exposure in the experiment. After UV exposure and releasing external force, why dose the shape change?
3. By incorporating photosensitizer, the hydrogel robot can be actuated by NIR irradiation with higher photo-thermal efficiency. Considering that NIR light should be focused on the specific area to move robot during motions, the spatial resolution of infrared radiation is crucial. The related parameters

should be provided in the manuscript.

4. In practical applications, hydrogel robots may face complex mechanical environments affecting the actuation performance, such as scratches, cuts, and ruptures. By introducing dynamic disulfide bonds, self-healing soft robots can be developed. Please clarify the formation and breaking mechanism of disulfide bonds.

5. Would the hydrogel robot perform reversible shape morphing in non-water environment?

6. The authors mentioned the effects of surrounding environments on actuation performance of hydrogel robots. There are some recent works focusing on excellent environmentally compatible hydrogel devices worth to consider in the article (*Advanced Functional Material*, 2021, 31, 2101464; *Small*, 2021, 17, 2101151).

We thank all the reviewers for their interests and rational concerns towards this work, which help us to improve the manuscript. Accordingly, this manuscript has been revised to clarify these issues in the main text and supporting information. Point-by-point responses to the reviewers are stated below.

Reviewer #1 (Remarks to the Author):

High actuation speed and reconfigurable actuation modes are two desired characteristics for soft hydrogel robots. Despite recent progresses, material systems exhibiting both are still under-explored. This manuscript presents a photothermally responsive hydrogel with rapid motions for which actuation modes can be programmed via dynamic exchange of the disulfide crosslinking. The actuation performances are impressive. In contrast to common mass-diffusion-based mechanism for previous hydrogel actuators, the manuscript demonstrated a unique mechanism for which the actuation relies mainly on conformational change of the anisotropic polymer chains after programming. Therefore, I would like to recommend its publication in Nature Communications after certain revisions by addressing the following issues.

1. It is not clear why the actuation speed of previous hydrogels with re-programmability (ref. 42) is slow. Please elaborate.

Author response: Thanks for this interesting question. The probable reason that constrains the actuation speed in ref. 42 (ref. 44 in the current version) is attributed to its hydrophobic domain in the system. Specifically, the poly(*N*-isopropylacrylamide) (PNIPAM) hydrogel in ref. 42 contains hydrophobic stearyl side chains which are applied to program the network anisotropy utilizing their hydrophobic aggregation. When triggering the actuation upon heating, the programmed PNIPAM chains collapse and tend to aggregate around the stearyl domains. This step is fast, which can be achieved within 2 min. Upon cooling, however, the aggregated PNIPAM chains suffer from slow dynamics to get away from the stearyl domains due to the strong hydrophobic interaction. As such, the conformation change of PNIPAM is slow in the cooling step, and the entire actuation cycle requires tens of minutes dominated by the cooling-induced actuation. In our current manuscript, there is no additional hydrophobic component to form strong interactions with PNIPAM, thus the conformation change of PNIPAM chains and the corresponding actuation speed of a whole cycle are both very fast.

The following statement has been added on page 3 in the manuscript.

“However, completing an actuation cycle requires tens of minutes dominated by the slow cooling-induced actuation step. Specifically, the collapsed PNIPAM chains tend to aggregate around the hydrophobic domains, and the conformation change upon cooling would be still slow since the strong hydrophobic interaction remarkably retard the chains from reswelling.”

2. It is expected that there is a gradient on the chain orientation, as reflected in the shape retention. However, the evidence is somewhat insufficient. More investigations concerning the gradient should be conducted.

Author response: We agree with the reviewer. In the revision, we use a lamination method to investigate the gradient distribution. Three identical hydrogel sheets with a thickness of 0.1

mm were overlaid, and the integrated sample was programmed (folded and irradiated) through the same approach as the common non-laminated sample. The angles of the three sheets after the programming were respectively 41° , 95° , 115° from top to bottom (Fig. S3), indicating a decreasing trend of shape retention because of the light attenuation along the thickness direction. The experimental results support the proposed mechanism of the gradient on chain orientation.

The above section has been added on page 5 in the manuscript.

Fig. S3. Programming of laminated samples. **a**, Scheme of the programming process. **b**, Photographs of the samples after programming from top to bottom. Scale bars are 0.5 cm.

3. It is known that two-way shape memory polymers and liquid crystal elastomers can also provide programmable actuations. Apart from the distinct mechanism, advantages on material performances of the presented hydrogels should be discussed.

Author response: Thanks for the good suggestion. Actuation of two-way shape memory polymers relies on the melting/formation of crystalline domains which exhibit large thermo-hysteresis within a heating-cooling cycle. Therefore, the actuation temperature range is large and the actuation speed is thus largely compromised. For liquid crystal elastomers, achieving the actuation programmability and reducing their actuation temperature to around body temperature are both very challenging, since they require highly sophisticated chemical design. In contrast, the presented hydrogel system provides a facile and inspiring way to realize the programmability in a very mild condition as well as fast actuation in a narrow temperature variation. Moreover, the water-abundant nature of hydrogels promises their biocompatibility, tissue-like modulus, and unique mass-transfer properties (for drug delivery), which are desired for many biomedical-related applications. The above discussion has been added on page 12 in the revision.

4. More characterizations (e.g. tensile tests) are required to show the material properties after doping the carbon particles.

Author response: Thanks for the suggestion. Tensile test of the carbon-doping PNIPAM hydrogel was conducted as shown in Fig. S11. The doping didn't result in remarkable alteration on the mechanical properties of the hydrogel.

Fig. S11. Mechanical property of the carbon-doped hydrogel.

Furthermore, the gradient distribution after the carbon doping was also investigated. The angles of the three sheets after the laminated programming were respectively 41° , 137° , 155° from top to bottom, indicating a larger gradient distribution because the carbon particles could enhance the light absorption.

Fig. S12. Gradient distribution of the carbon-doped samples. The experiment method is the same as that in Fig. S2.

The following description has been added on page 14 in the revision.

“The doping didn’t result in remarkable alteration on the mechanical properties of the hydrogel (Fig. S11), but introduced a larger stress gradient along the thickness due to the enhanced light absorption of carbon doping (Fig. S12). The shape retention reduced to $53.2\pm 2.2\%$, and the actuation angle slightly decreased to $74.7\pm 4.2^\circ$ (Fig. S13).”

Reviewer #2 (Remarks to the Author):

In this paper, the authors developed a soft underwater robot based on thermal-responsive hydrogel crosslinked by a disulfide crosslinker (BISS). Owing to the photo-responsive dynamic bonds in hydrogel network, the robot allows photo-mechanical programming to introduce diverse motion modes. The factors affecting the shape reconfiguration are explored and the optimal parameters for best actuation performance are chosen. Moreover, the authors also compared the speed of conventional actuation mechanism based on water diffusion and this actuation mechanism. Based on the conformation reconfiguration of the polymer chains, the hydrogel robot realizes fast actuation (1.7 Hz), with great potential for fabrication of advanced hydrogel devices in the future. In general, this work is interesting and clearly organized. However, there are still some details remaining to be improved before acceptance.

1. The authors introduced the actuation mechanism of this hydrogel robot and mentioned that the stress of a uniaxial stretched hydrogel sample will be relaxed through photo-mechanical programming via disulfide exchange. Why dose UV irradiation result in disulfide exchange? To make it clear, the reason should be added.

Author response: Thanks for the question. Disulfide is a typical dynamic covalent bond that would exchange under some specific stimuli (e.g. heat, light, redox). Here in this system, UV source is absorbed by a photo-initiator (I2959) to produce radicals. These radicals attack and break the disulfide bonds to form thiyl radicals. The thiyl radicals will further be randomly recoupled into new disulfides. This break-recouple process results in the disulfide exchange. (*Macromolecules* 2011, 44, 8, 2444-2450). The explanation has been added on page 3 and the related reference has been cited as ref 45 in the manuscript.

2. During shape reconfiguration, the authors mentioned that external force should be applied under UV exposure. The reviewer is curious about how to apply external force under UV exposure in the experiment. After UV exposure and releasing external force, why dose the shape change?

Author response: For the first question, we applied external force on the sample with the aid of tools like glass sheets and clips for the fixing. Take the bending model (Fig. 2a) for example, the sample was folded into half and placed between two glass slides fixed by two clips (Fig. S6). For spatial irradiation, the unexposed areas were shielded by tin foils.

For the second question, the shape change is attributed to the rearrangement of the network topology based on disulfide bond exchange. When the sample is deformed into a certain strain, the polymer chains are stretched, and the entropy of the entire network decreases. Upon activation of the disulfide exchange via UV exposure when retaining the deformation, the polymer network can dissipate the stored energy through the topology rearrangement thus the system entropy increases again. As a result, the polymer network will not recover to the original state upon removal of the external force, and thus, the macroscopic shape changes after the UV programming (ref. 46-48). The explanation has been added on page 5 of the manuscript.

Fig. S6. Illustration showing the fixing of the bending deformation in Fig. 2a.

3. By incorporating photosensitizer, the hydrogel robot can be actuated by NIR irradiation with higher photo-thermal efficiency. Considering that NIR light should be focused on the specific area to move robot during motions, the spatial resolution of infrared radiation is crucial. The related parameters should be provided in the manuscript.

Author response: Thanks for the good suggestion. The relationship of the exposed area and distance between the light source and the sample was tested and presented in Fig. S14. Specifically, the irradiation area is 22 mm² at the distance of 4 cm that we employed to trigger the sample. Such an irradiation area fully covers the actuation region of our hydrogel sample.

More accurate control is not invested in the current work due to the lack of the light source but is worth exploring in the near future. The related statement has been added on page 14 in the revision.

Fig. S14. NIR-irradiation area with various distances between the light source and the sample. The irradiation area is calculated from the NIR spot area of three parallel tests.

4. In practical applications, hydrogel robots may face complex mechanical environments affecting the actuation performance, such as scratches, cuts, and ruptures. By introducing dynamic disulfide bonds, self-healing soft robots can be developed. Please clarify the formation and breaking mechanism of disulfide bonds.

Author response: Thanks for this inspiring suggestion. The mechanism for the disulfide exchange has been explained in the response to Comment 1. However, the samples do not provide observable self-healing performance after UV irradiation for 10 min as shown in the Figure below. It has been confirmed that the self-healing performance requires harsher conditions than network reconfiguration (ACS Appl. Mater. Interfaces 2017, 9, 22077-22082; Chem. Rev. 2021, 121, 1716-1745). In addition, the relatively high modulus of the hydrogels reflects the high crosslinking density, which reduces the mobility of the polymer chains and thus prevents the materials from healing (Polymer, 2013, 54, 6381-6388). In this case, we are planning to prepare self-healing soft robots based on the same mechanism but alternative systems by reducing the crosslinking density and changing the dynamic bonds with higher activity (e.g. imine bonds and acyl hydrazone).

The samples do not show observable self-healing performance after UV irradiation .

5. Would the hydrogel robot perform reversible shape morphing in non-water environment?

Author response: Yes. As shown in Figure 4c, the hydrogel can execute fast and reversible actuation in an oil environment which prohibits water exchange between the hydrogel and the environment. This fact strongly supports our hypothesis that the actuation of the hydrogel relies on thermoresponsive chain conformation change instead of the volumetric expansion/shrinkage due to the water exchange with the environment.

6. The authors mentioned the effects of surrounding environments on actuation performance of hydrogel robots. There are some recent works focusing on excellent environmentally compatible hydrogel devices worth to consider in the article (Advanced Functional Material, 2021, 31, 2101464; Small. 2021, 17, 2101151).

Author response: The related references have been added as ref 10 and ref 25.

REVIEWERS' COMMENTS

Reviewer #1 (Remarks to the Author):

I think the authors have revised the manuscript carefully by addressing the comments. So, I recommend its publication now.

Reviewer #2 (Remarks to the Author):

The authors have addressed all issues from the the reviewers and the submission is recommended for acceptance at the current stage.